# Large-Scale Integration of Amplicon Data Reveals Massive Diversity within *Saprospirales*, Mostly Originating from Saline Environments

**DOI:** 10.3390/microorganisms11071767

**Published:** 2023-07-06

**Authors:** Rafaila Nikola Mourgela, Antonios Kioukis, Mohsen Pourjam, Ilias Lagkouvardos

**Affiliations:** 1School of Chemical and Environmental Engineering, Technical University of Crete, 73100 Chania, Greece; rmourgela@tuc.gr; 2Department of Microbiology and Microbial Pathogenesis, School of Medicine, University of Crete, 71500 Heraklion, Greece; antokioukis@gmail.com; 3ZIEL Institute of Food and Health, Technical University of Munich, 85354 Freising, Germany; m.pourjam@tum.de; 4Institute of Marine Biology, Biotechnology and Aquaculture, Hellenic Centre for Marine Research, 71500 Heraklion, Greece

**Keywords:** *Saprospirales*, *Saprospiraceae*, bacterial diversity, integrative analysis, microbial dark matter, organic matter degradation, marine biodiversity, microbial ecology

## Abstract

The order *Saprospirales*, a group of bacteria involved in complex degradation pathways, comprises three officially described families: *Saprospiraceae*, *Lewinellaceae*, and *Haliscomenobacteraceae*. These collectively contain 17 genera and 31 species. The current knowledge on *Saprospirales* diversity is the product of traditional isolation methods, with the inherited limitations of culture-based approaches. This study utilized the extensive information available in public sequence repositories combined with recent analytical tools to evaluate the global evidence-based diversity of the *Saprospirales* order. Our analysis resulted in 1183 novel molecular families, 15,033 novel molecular genera, and 188 K novel molecular species. Of those, 7 novel families, 464 novel genera, and 1565 species appeared in abundances at ≥0.1%. *Saprospirales* were detected in various environments, such as saline water, freshwater, soil, various hosts, wastewater treatment plants, and other bioreactors. Overall, saline water was the environment showing the highest prevalence of *Saprospirales*, with bioreactors and wastewater treatment plants being the environments where they occurred with the highest abundance. *Lewinellaceae* was the family containing the majority of the most prevalent species detected, while *Saprospiraceae* was the family with the majority of the most abundant species found. This analysis should prime researchers to further explore, in a more targeted way, the *Saprospirales* proportion of microbial dark matter.

## 1. Introduction

In the last two decades, numerous studies have reported the presence of bacteria belonging to the order of *Saprospirales* in various artificial and natural ecosystems. Bacteria belonging to *Saprospirales* can be found in natural ecosystems such as marine and freshwater environments, and soils [1,2,3,4,5,6,7,8,9,10,11,12,13,14,15,16,17,18,19,20,21,22]. Furthermore, several studies have reported that *Saprospirales* members can be found in wastewater treatment plants, and especially in activated sludge and anaerobic ammonium oxidation reactors [23,24,25,26,27,28,29,30,31,32,33,34,35]. Regarding the latter, hydrolysis of proteins, denitrification, aromatic compound degradation, and organic matter decomposition are confirmed capabilities of certain bacteria belonging to the order *Saprospirales* [30,32,33,34,35,36,37].

The order Saprospirales belongs to the class of Saprospiria and comprises three families: Saprospiraceae, Lewinellaceae, and Haliscomenobacteraceae [38]. Both Lewinellaceae and Haliscomenobacteraceae comprise three genera: Flavilitoribacter, Neolewinella, and Lewinella; and Haliscomenobacter, Phaeodactylibacter, and Portibacter, respectively [3,6,7,8,9,10,11,15,17,22,24,38,39,40]. On the other hand, Saprospiraceae comprises four valid published genera: Aureispira, Membranihabitans, Rubidimonas, and Saprospira, and seven candidate genera: Candidatus Aquirestis, Candidatus Epiflobacter, Candidatus Brachybacter, Candidatus Opimibacter, Candidatus Parvibacillus, Candidatus Defluviibacterium, and Candidatus Vicinibacter [12,13,14,16,20,21,23,26,41,42]. As far as the classified species are concerned, the genus of Flavilitoribacter contains Flavilitoribacter nigricans, the genus of Lewinella contains Lewinella cohaerens, and Neolewinella contains twelve species: Neolewinella agarilytica, Neolewinella antarctica, Neolewinella aquimaris, Neolewinella aurantiaca, Neolewinella lacunae, Neolewinella litorea, Neolewinella lutea, Neolewinella marina, Neolewinella maritima, Neolewinella persica, and Neolewinella xylanilytica [3,6,7,8,9,10,11,38,39,40,43]. Portibacter contains Portibacter lacus and Portibacter marinus, Haliscomenobacter contains Candidatus Haliscomenobacter calcifugiens and Haliscomenobacter hydrossis, and Phaeodactylibacter contains Phaeodactylibacter luteus and Phaeodactylibacter xiamenensis [12,15,17,22,24,40]. Regarding the genera belonging to the family of Saprospiraceae, Aureispira contains two classified species: Aureispira marina and Aureispira maritima [13,14]. Each of the genera Membranihabitans, Rubidimonas and Saprospira contain one classified species: Membranihabitans marinus, Rubidimonas crustatorum, and Saprospira grandis, respectively [16,21,41,42]. The genus of Candidatus Aquirestis contains Candidatus Aquirestis calciphila, while the genus of Candidatus Epifloribacter does not contain any classified species [12,23]. Finally, Candidatus Brachybacter algidus, Candidatus Opimibacter skivensis, Candidatus Opimibacter iunctus, Candidatus Parvibacillus calidus, Candidatus Defluviibacterium haderslevense, Candidatus Vicinibacter proximus and Candidatus Vicinibacter affinis were proposed by Kondrotaite et al. [26].

The diversity of *Saprospirales* has been described using traditional isolation methods. As with most microbial taxa, it is expected that a higher number of undescribed species exist within *Saprospirales* that are simply uncultivable with our current protocols. In recent decades, the advances in next-generation sequencing have led to the production of vast numbers of sequences that have accumulated in public repositories. In total, this integrated information represents a thorough global sampling effort. Databases, such as IMNGS, contain 16S rRNA microbial profiles from more than 500,000 pre-processed samples across the globe. In this study, the extensive information provided by public repositories was combined with recent analytical tools to evaluate the global diversity of the microbial order *Saprospirales.* The predicted novel sequence types, combined with environmental origin metadata, allowed for the first global assessment of the ecology and diversity of the order in question.

## 2. Materials and Methods

To create the initial dataset of sequences associated with the order of *Saprospirales*, we executed taxonomy and similarity queries in the IMNGS and SILVA databases [44,45]. As both of the aforementioned databases are not up to date with the recent splitting of the *Saprospirales* order, the search term was limited to “*Saprospiraceae*”. Regarding the taxonomy query, sequences classified as the family of interest were extracted from SILVA. These sequences were also used as the input in IMNGS for the similarity query. The gathered sequences (*n* = 988 K) were dereplicated, and then aligned and reclassified using SINA (v.1.7.2) with SILVA SSU database (v.138) [46]. Reclassification took place to update the taxonomy information to retain the sequences belonging to the *Saprospirales* order. Following this, the novel tool “taxonomy informed clustering” (TIC) [47] was used to process the sequencing data. TIC is a new clustering algorithm that first procedurally divides taxonomically annotated sequences into bins of the same taxonomy down to the genus level. Then, it performs incremental clustering using the sequences confined within the same taxonomy level to avoid contamination of the clusters with sequences with clearly different phylogenetic origin but otherwise overall sequence similarity above the set cut-off levels.

Since different studies use different sequencing technologies or different primers, the resulting 16S rRNA gene fragments do not always overlap. To identify the most represented region of the 16S rRNA gene in our dataset, we calculated the representation of each position in the SINA multiple sequence alignment as the sum of all bases in that position. After identification of the most represented region, all sequences were trimmed around these positions. The sequences with more than 80% of the number of bases that aligned with those *Escherichia coli* would have in this region were selected for further analysis. All sequences that did not cover or partially covered that region were removed.

The 16S reference sequences of 24 known Saprospirales species were obtained directly from the NCBI and the SILVA databases. For the 7 remaining described species, their 16S rRNA gene sequence was extracted from their respective genomes using BLAST, with the 16S reference sequence of *Aureispira Maritima* as a query. Finally, the 31 reference 16S sequences were aligned and trimmed around the selected SINA positions.

Traditionally, the cut-offs of 97%, 95%, and 90% 16S rRNA gene similarity are used to denote sufficient evolutionary divergence for the classification of distinct species, genera, and families respectively. Nevertheless, smaller regions of the 16S rRNA gene do not always mirror the evolutionary information captured by the whole gene. Adjusting similarity cut-offs for selected regions is important to avoid over or underestimation of diversity. For the selected region of the 16S rRNA gene, we evaluated the corresponding similarity cut-offs to be used for clustering of species, genera, and families, based on the actual sequence distances among all known *Saprospirales* species. We found that the existing known species, when compared over the selected region, showed 96% similarity among species of the same genus, 92% similarity among species across genera of the same family, and 90% similarity among species belonging to different families, on average. Those values were used as clustering cut-offs in TIC for determining the diversity of molecular species (sOTUs), molecular genera (gOTUs), and molecular families (fOTUs), respectively.

For the ecological analysis, the metadata for each sequence in our dataset, available in the IMNGS database, was used. We extracted information from IMNGS related to the environment where each molecular species (sOTU) was detected (prevalence), as well as their abundances in each of those samples. The ecological analysis was focused on the sOTUs with an abundance of ≥0.1% in at least one sample. Samples with unclear origins were manually determined by following their sequence read archive (SRA) accession numbers in SRA site (https://www.ncbi.nlm.nih.gov/sra, Accessed during January 2023). Manually assembled systems, such as laboratory-cultivated photosynthetic mats and biofilms on polymer material surfaces, were removed from the ecological analysis. We considered only the samples derived from natural environments as well as artificial environments such as wastewater treatment plants. Briefly, the natural environments in which the *Saprospirales* species were detected were saline water, saline water sediments, and beach sand (hereafter referred to as “saline water”), freshwater and freshwater sediments (referred to as “freshwater”), soil, air, terrestrial flora and fauna (referred as “plant” and “host”, respectively), as well as saline water flora and fauna (referred as “plant saline water” and “host saline water”, respectively). In addition, *Saprospirales* species were detected in samples derived from wastewater treatment plants, bioreactors, activated sludges, and fermentation processes (referred to as a “bioreactor”).

Finally, known species were assigned to formed sOTUs when BLAST similarity was above 98%. Some species were not distinguishable in the selected region as they were assigned to the same sOTU. In the text, those undistinguishable sOTUs carry both the names of the closest known species, i.e., *Neolewinella marina/litorea*, *Neolewinella persica/agarilytica*, *Portibacter lacus/marinus*, and *Vicinibacter affinis/proximus*.

The general workflow that was followed (Figure 1) can be easily adapted to other taxonomic groups of interest, simplifying future microbial diversity studies.

## 3. Results

### 3.1. Processing Results

The evaluation of the most represented region of 16S rRNA gene sequences in our Saprospirales dataset pointed to a region spanning the 10 K to 25.5 K positions of the complete SINA alignment (Figure 2). Our estimation of expected bases across this region, using the SINA-aligned 16S rRNA gene of *E. coli*, was 282 bases. Following our requirement for an 80% minimum coverage over this region, we exclude every sequence that had less than 229 bases around our selected positions. After eliminating sequences targeting different regions, our final dataset was limited to 691 K, from 988 K sequences initially.

The incremental taxonomy bounded clustering of the final sequencing data with TIC resulted in 118 K novel molecular species (sOTUs), 9 K molecular genera (gOTUs), and 1269 families (fOTUs) (A FASTA formatted file “Saprospirales_diversity.zip” with all the sOTUs and their assigned taxonomy is available in the Appendix A). Characterization of species, genera, and families, when referring to the results of our analysis, should always be considered as short forms of “clusters of sequences with similarities over the selected region equivalent to the corresponding taxonomic level”. Therefore, our sOTUs, gOTUs, and fOTUs are not equivalent to official taxonomies. The sequences clustered in one sOTU at 96% similarity could belong to multiple biological species. This means that our method tended to underestimate the diversity compared to how common practices (ANI, phylogeny, function) would determine how the diversity of biological species is assigned to all isolates carrying the variants of the 16S rRNA gene in a dataset.

### 3.2. Analysis of Results

There were 204 K samples, out of 500 K pre-processed samples in the IMNGS database, that covered our selected region of interest. Within those, *Saprospirales* were found in almost 13% of IMNGS samples (Figure 3). Specifically, 48% of freshwater samples, 33% of saline water samples, 26% of plant samples, 22% of soil-derived samples, and 1% of samples were found to be positive, originating from hosts (terrestrial fauna). Furthermore, it was found that the 13% of samples marked as “other” were positive, which included samples derived from different environments such as wastewater treatment plants, bioreactors, activated sludges, fermentation processes, air, and saline water flora and fauna.

Regarding the predicted number of molecular families, genera, and species belonging to the order of *Saprospirales*, a small percentage of species appeared in abundance at ≥0.1% (Table 1). Only nine families had species with an abundances of ≥0.1%, including the three known families, meaning that only the 0.71% of the predicted families had at least one species with an abundance of ≥0.1%. Similarly, 479 genera (3.18% of the predicted genera) had at least one species with an abundance of ≥0.1%, including all known genera except for *Rubidimonas* and *Saprospira*. Overall, 1565 species (1.33% of the predicted species) presented an abundance of ≥0.1%.

To be more specific, and considering only sOTUs with an abundance of ≥0.1%, they were distributed in 132 genera belonging to *Saprospiraceae*, including 9 out of 11 known genera in the family; in 186 genera belonging to *Lewinellaceae*; and 150 genera to *Haliscomenobacteraceae*, including all known genera to both families (Table 2). The remaining seven unknown families had 1 to 4 genera, resulting in 11 genera with a species abundance of ≥0.1%. Overall, regardless of the species abundance, most of the predicted genera were classified as *Lewinellaceae.*

Regardless of species abundance, almost all predicted species were classified to known families (Table 3). Specifically, 98% of the predicted species and 99% of species with abundances of ≥0.1% were classified as the three known families. Concerning these two groups, most of the species classified to *Saprospiraceae* belonged to known genera (53% and 56% with a ≥0.1% species abundance, respectively), while most of the species classified to *Lewinellaceae* and to *Haliscomenobacteraceae* belonged to unknown genera (56% and 59% with ≥0.1% species abundance, and 80% and 68% with ≥0.1% species abundance, respectively) (Table 3 and Table 4).

In particular, almost 82% of the unknown families had only one species, while 0.17% of the unknown families had more than 51 species classified to each of them (Figure 4). Furthermore, almost 17% of the unknown families had 2 to 10 species, while the rest had 11 to 50 species. On the other hand, all unknown families with a species abundance of ≥0.1% had less than 10 species. To be more specific, these seven unknown families had one to six species each. Five families had one species, one family had two species and the last unknown family had six species classified to it. Moreover, the aforementioned family had five genera, while the rest of these families had only one. Regarding the known families, the majority of the unknown genera within the three known families had 2 to 10 species each, regardless of their abundance (Figure 4). The latter also applies to the known genera of these families, except for *Saprospiraceae* species with an abundance of ≥0.1%. In this case, most of the known genera had either 2 to 10 species or at least 51 species.

Regarding the environmental distribution of *Saprospirales* families, saline water was the environment where most species were present and had their maximum abundance (Figure 5). Briefly, for *Haliscomenobacteraceae* the descending order of environments according to species prevalence was saline water > host saline water > freshwater > bioreactor > plant saline water > soil > air > host. Similarly, for *Lewinellaceae* the corresponding descending order was saline water > bioreactor > host saline water > freshwater > soil > plant saline water > air = host, and for *Saprospiraceae*, it was saline water > bioreactor > freshwater > soil > host saline water > host > plant saline water. As far as the environments with the maximum species abundance for each known family are concerned, the respective descending orders were: for *Haliscomenobacteraceae*, saline water > host saline water > freshwater > bioreactor > plant saline water > soil > air > host > plant, for *Lewinellaceae*, saline water > bioreactor > freshwater > host saline water > soil > plant saline water > host > air = plant, and for *Saprospiraceae*, saline water > bioreactor > freshwater > soil > host saline water > plant saline water > host > plant. Collectively, concerning the unknown families (7 families containing 11 species), the descending order of environments according to species prevalence and maximum abundance was saline water > plant saline water > soil > host saline water = host.

As far as the known genera are concerned, the most prevalent environments, as well as the maximum abundance environments with the greatest number of species, were found to be different for each genus (Figure 6). It is noted that the number of species belonging to the known genera and with abundances of ≥0.1% was 681 (Table 3). For *Aquirestis,* the descending order of environments according to the number of species detected in each environment was freshwater > saline water > bioreactor > soil = host. For *Aureispira,* the corresponding descending order was saline water > host saline water > soil = freshwater. For *Brachybacter,* the descending order according to species maximum abundance environment was bioreactor > saline water > soil > host saline water > freshwater = host = plant saline water, while the corresponding descending order according to species prevalence was almost the same, i.e., bioreactor > saline water > soil > host saline water > freshwater = host. For *Defluviibacterium*, the respective order was bioreactor > freshwater. For *Membranihabitans,* the descending order of environments was host saline water > soil > saline water = bioreactor > host. For *Epiflobacter*, the order was freshwater > saline water = bioreactor, whilst for *Opimibacter,* the order was soil > saline water > bioreactor > host saline water > freshwater > host. Concluding for *Saprospiraceae*, for *Vicinibacter* species, the descending order of environments according to their prevalence was bioreactor > saline water > plant saline water > soil > freshwater, whilst for maximum abundance environments, the corresponding order was almost the same, i.e., bioreactor > saline water > plant saline water > soil. Regarding the family of *Lewinellaceae*, for *Lewinella* species, the descending order of environments according to their maximum abundance was saline water > freshwater = host saline water > plant saline water > bioreactor > air, while according to their prevalence, the order was saline water > plant saline water > freshwater > host saline water > bioreactor > air. For *Neolewinella,* the descending order of environments regarding species prevalence was saline water > plant saline water > freshwater > host saline water > bioreactor, while for maximum abundance environments, the corresponding order was saline water > plant saline water > host saline water > freshwater > bioreactor > air. Similarly, for *Flavilitoribacter*, the orders of environments were saline water > host saline water > bioreactor = freshwater = soil > host, and saline water > host saline water > bioreactor = freshwater > soil > host. Regarding the family of *Haliscomenobacteraceae*, for *Haliscomenobacter* species, the descending order of environments according to their prevalence was bioreactor = freshwater > saline water > host saline water = soil = air, while according to their maximum abundance, the order was bioreactor > freshwater = saline water > host saline water = soil = air. For *Phaeodactylibacter,* the corresponding descending order of environments was saline water > host saline water > freshwater > bioreactor > soil = host. Finally, for *Portibacter,* the order of environments was saline water > host saline water > plant saline water > freshwater.

As far as the unknown genera are concerned, regardless of their classification in terms of families, saline water was the environment where most species of these genera were present and had their maximum abundance (Figure 7). It is noted that, for species abundance of ≥0.1%, the total number of unknown genera belonging to the three known families, as well as to the unknown families, was 462; the species belonging to them were 884 (Table 2 and Table 3). To be more specific regarding their environmental distribution, 53.24% of the species belonging to these unknown genera were found in saline water, and 51.33% of the species had their maximum abundance in samples derived from saline water. On the other hand, the remaining environments appeared in smaller percentages, i.e., freshwater, 12.27% and 12.17%; bioreactor, 12.73% and 13.33%; saline water hosts, 11.11% and 12.05%; soil, 6.02% and 5.91%; saline water plants, 3.47% and 3.71%; air, 0.58%; and terrestrial hosts, 0.58% and 0.70%, respectively. Furthermore, 0.23% of species had their maximum abundance in samples derived from plants.

Cosmopolitan species, i.e., species that appeared at least in 50 samples, constituted almost 16% of species with an abundance of ≥0.1%, i.e., 244 species out of 1565 species. Most of the cosmopolitan species showed maximum abundances between 0.1% and 1%, while fewer cosmopolitan species showed maximum abundances between 1% and 5%, and even fewer species showed maximum abundances above 5% (Figure 8). Specifically, eight species showed maximum abundances above 5%, four of which had abundances above 10%. As mentioned previously, saline water was both the most prevalent environment and the environment where the majority of species had their maximum abundance, followed by environments related to bioreactors and freshwater. Further analysing the cosmopolitan species, almost all species belonged to known families, except one species that belonged to an unknown family. Furthermore, 97 species belonged to *Saprospiraceae*, 32 of which belonged to unknown genera; 85 species belonged to *Lewinellaceae*, 31 of which belonged to unknown genera; and 61 species belonged to *Haliscomenobacteraceae*, 32 of which belonged to unknown genera.

The known genera that correspond to these cosmopolitan species showed environmental preferences, unlike unknown genera that had species detected in a variety of environments (Figure 8). Specifically, *Aureispira* showed an environmental preference for saline water; *Aquirestis* showed an environmental preference for freshwater; *Opimibacter,* for soil; *Brachybacter* and *Vicinibacter*, for bioreactor-related environments; and *Membranihabitans*, for terrestrial hosts. *Defluviibacterium* had few cosmopolitan species, and almost all of them were detected in bioreactors and wastewater treatment plants. In addition, *Epiflobacter* was detected solely in freshwater. *Parvibacillus* was equally detected in saline water and bioreactor-related environments. On the other hand, unknown genera belonging to *Saprospiraceae* could be found in saline water, soil, freshwater, hosts, bioreactors, and wastewater treatment plants. Regarding the *Lewinellaceae* family, *Neolewinella*, *Lewinella*, and *Flavilitoribacter* showed environmental specificity in saline water, while unknown genera could be found in saline water, freshwater, bioreactors, and wastewater treatment plants. Regarding *Haliscomenobacteraceae*, both *Portibacter* and *Phaeodactylibacter* showed an environmental preference for saline water, while *Haliscomenobacter* was found in freshwater, bioreactors, and wastewater treatment plants. *Haliscomenobacteraceae* unknown genera could be detected in saline water, freshwater, bioreactors, and wastewater treatment plants. Finally, the one species that belonged to the unknown family was found in freshwater.

Twenty-eight known species were represented by twenty-four sOTUs clusters (Table 5). After further analysing these known species, the number of positive samples with a species abundance of ≥0.1% ranged from 1 to 282 samples; in addition, their maximum abundances ranged from 0.11% to 4.84%. Saline water was the most prevalent environment, as well as the environment where most of these species had their maximum abundance. (Table 5 and Figure 9). In general, *Brachybacter algidus*, *Defluviibacterium haderslevense*, *Parvibacillus calidus*, and *Vicinibacter affinis/Proximus*, all of which belong to the family of *Saprospiraceae*, were found in samples that originated from wastewater treatment plants and bioreactors. *Aureispira marina*, *Aureispira maritima*, and *Membranihabitans marinus* were found in saline-water-related environments, while *Aquirestis calciphila* was found exclusively in freshwater samples. Species belonging to the genus of *Epiflobacter* were found in freshwater and saline water hosts. In contrast, *Opimibacter skivensis* was found in all detected environments, i.e., soil, freshwater, bioreactors, plants, and saline water. *Lewinella cohaerens* and *Flavilitoribacter nigricans* appeared in a variety of unrelated environments, unlike the rest of the *Lewinellaceae* species, which were found only in saline-water-related environments. In particular, both *Lewinella cohaerens* and *Flavilitoribacter nigricans* were found in samples related to bioreactors, and activated sludge, freshwater, and saline water. Both species were more prevalent in samples originating from saline water and had their maximum abundance in samples from a nitrifying bioreactor and activated sludge, respectively. On the other hand, all known *Haliscomenobacteraceae* species appeared either in saline water (*Portibacter lacus/marinus*, *Haliscomenobacter hydrossis* and *Phaeodactylibacter xiamenensis)* or in freshwater *(Phaeodactylibacter luteus)*. *Opimibacter iunctus*, *Rubidimonas crustatorum*, *Saprospira grandis*, *Haliscomenobacter calcifugiens*, and *Neolewinella lacunae* were not detected above a 0.1% abundance in any sample, therefore they lacked environmental distribution.

The prevalence of known species was not uniform across the environments, indicating varying levels of niche specificity. For example, *Aquirestis calciphila* and *Vicinibacter affinis/proximus* appeared in five samples, all of which were derived from freshwater, and bioreactor and wastewater treatment processes, respectively. *Aureispira marina* and *Aureispira maritima* were found in six and seven samples, respectively, originating from saline-water-related environments, specifically saline water, and saline water flora and fauna. *Epiflobacter* species were found in six samples, the majority of which originated from freshwater-related environments. In addition, *Membranihabitans marinus* appeared in fifteen samples derived from saline water hosts. Further, *Opimibacter skivensis* was detected mainly in soil (211 out of 282 samples). It was also found in bioreactors, plants, freshwater, and saline water. Also, *Neolewinella xylanilytica*, *Neolewinella aquimaris*, *Neolewinella marina/litorea,* and *Neolewinella lutea* appeared in seven to nine samples, all of which were derived from saline-water-related environments. *Lewinella cohaerens* and *Flavilitoribacter nigricans* were found in ten and eight samples, respectively, the majority of which were derived from saline water.

*Opimibacter skivensis* was also the most prevalent species in this study, as it appeared in 3066 samples, 282 of which had abundances of ≥0.1% (Table 6). Moreover, almost 75% of samples were derived from the soil. *Opimibacter skivensis* appeared to be low in abundance, as its maximum abundance was 1.53%. *Flavilitoribacter nigricans* and *Neolewinella aquimaris* were also among the top 20 most prevalent species, which appeared in 315 samples (8 samples with ≥0.1% species abundance) and 256 samples (9 samples with ≥0.1% species abundance), respectively. The dominant environment for both species was saline water. On the other hand, *Flavilitoribacter nigricans* was found to be most abundant in bioreactor processes, while *Neolewinella aquimaris* was most abundant in saline water. Unlike *Opimibacter skivensis*, both *Flavilitoribacter nigricans* and *Neolewinella aquimaris* had low maximum abundances, specifically 0.22% and 0.33%, respectively. Overall, for the top 20 most prevalent species, the most prevalent environment, and the environment where most species had their maximum abundance was saline water (Table 6).

Unlike the top 20 most prevalent species, all of the top 20 most abundant species remained unidentified (Table 7). Species abundance ranged from 5.89% to 28.02%. In addition, the most prevalent environments were saline water and bioreactor processes, while the environment where most species had their maximum abundance detected was in samples derived from bioreactor processes. To sum up, saline water may be the environment with the highest prevalence, but bioreactors and wastewater-treatment-related processes had the most abundance (Table 6 and Table 7). Finally, *Lewinellaceae* is the family that contained the majority of the most prevalent species (Table 6), while *Saprospiraceae* is the family that contained the majority of the most abundant species (Table 7).

## 4. Discussion

*Saprospirales* can indeed be found in various environments, such as saline water, freshwater, soil, bioreactors, and wastewater treatment plants, as has been already mentioned in various studies [1,2,3,4,5,6,7,8,9,10,11,12,13,14,15,16,17,18,19,20,21,22,23,24,25,26,27,28,29,30,31,32,33,34,35]. In addition, air, plants, as well as terrestrial and saline water hosts are also potential environments for *Saprospirales.* In this work, we present that saline water was the environment that *Saprospirales* were most prevalent in, but bioreactors and wastewater treatment plants were the environments in which they were found in the highest abundance. Furthermore, *Lewinellaceae* is the family that contains the majority of the most prevalent species, while *Saprospiraceae* is the family that contains the majority of the most abundant species.

Regarding cosmopolitan species, i.e., species that appeared at least in 50 samples, saline water was the environment with the highest prevalence, where the majority of species had their maximum abundance, followed by bioreactor-related environments and freshwater. In general, almost all known genera that belong to *Saprospiraceae*, *Lewinellaceae*, and *Haliscomenobacteraceae* showed environmental preferences that corresponded with already published studies. For instance, *Aquirestis* showed an environmental preference for freshwater [12]; *Aureispira* showed an environmental preference for saline water [13,14]; and *Brachybacter*, *Defluviibacterium*, and *Vicinibacter* showed environmental specificity in bioreactors and wastewater treatment plants [26]. Furthermore, *Neolewinella*, *Lewinella*, *Flavilitoribacter*, *Portibacter,* and *Phaeodactylibacter* showed environmental specificity in saline water [3,7,8,9,10,11,15,17,22,43]. *Haliscomenobacter* appeared in saline water, freshwater, bioreactors, and wastewater treatment plants, which agrees with the previous studies that associate this genus with wastewater treatment plants and freshwater environments. [12,24,25]. On the other hand, *Membranihabitans* showed environmental specificity in terrestrial hosts, which opposes the findings of previous studies; Li et al. [16], and Béziat et al., found *Membranihabitans* in saline water environments. *Opimibacter* showed environmental specificity in soil. *Parvibacillus* appeared equally in saline water and activated sludge, while Kondrotaite et al. [26] related these with the wastewater treatment plants. Finally, *Epiflobacter* appeared in freshwater, while Xia et al. [23] associated this genus with activated sludge.

Twenty-eight known species were represented by twenty-four sOTUs clusters (≥ 0.1% species abundance). Therefore, not all described species could be detected in this integrated analysis as abundant constituents of microbial communities across the more than 200 K samples tested. If this observation was extended to the 1565 species-level clusters formed that were detected in abundances of ≥0.1%, we could safely and conservatively estimate that the global *Saprospirales* diversity is at least 2000 species. Concerning the niche assignment of known species, it agrees to a high extent with that of already published studies. Specifically, concerning *Saprospiraceae*, *Aquirestis calciphila* was found in freshwater samples [12], and *Aureispira marina* and *Aureispira maritima* were found in saline-water-related environments [13,14]. In particular, *Aureispira maritima* was found to be most prevalent in saline water hosts and most abundant in saline-water-derived samples, while *Aureispira marina* had both a prevalence and maximum abundance in saline water. *Brachybacter algidus*, *Defluviibacterium haderslevense*, *Parvibacillus calidus*, and *Vicinibacter affinis/proximus* were found in samples that originated from wastewater treatment plants and bioreactors, as reported by Kondrotaite et al. [26]. *Membranihabitans marinus* was found in saline water hosts, which was in alignment with the studies of Li et al. [16] and Béziat et al. [49]. On the other hand, *Opimibacter skivensis* was found in a variety of environments, especially in soil samples. Kondrotaite et al. [26] reported that this species should be found in wastewater treatment plants. *Opimibacter skivensis* was found in 282 samples with an abundance of ≥0.1%, 211 samples of which were derived from soils, while 18 samples were derived from bioreactor-related and wastewater-treatment-related samples. It was also found in plants, hosts, freshwater, and saline water. Regarding *Haliscomenobacteraceae*, both *Phaeodactylibacter xiamenensis*, and *Portibacter lacus* were found in saline water samples, as already published studies indicate [17,22]. Contrary to the aforementioned information, we detected *Phaeodactylibacter luteus* in freshwater, while Lei et al. [15] isolated *Picochlorum sp*. from saline water alga, and Ma et al. [29] found it in wastewater-related environments. *Haliscomenobacter hydrossis* was detected in saline water environments, while it has been reported that this species is usually found in activated sludge [24,25]. Finally, regarding *Lewinellaceae*, all species appeared to be most prevalent and most abundant in saline water environments, except *Lewinella cohaerens* and *Flavilitoribacter nigricans*. These had maximum abundances in samples from a nitrifying bioreactor and activated sludge, respectively, despite being more prevalent in samples originating from saline water. However, Khan et al. [43] associated both species with marine environments. Unlike the rest of the *Lewinellaceae* species, both *Lewinella cohaerens* and *Flavilitoribacter nigricans* appeared in a variety of unrelated environments, i.e., in samples related to bioreactors and activated sludge, freshwater, and saline water. Regarding the remaining *Lewinellaceae* species, our results conformed with those of previous studies. *Neolewinella persica/agarilytica*, *Neoewinella antarctica*, *Neolewinella aquimaris*, *Neolewinella aurantiaca*, *Neolewinella lutea*, *Neolewinella marina/litorea*, *Neolewinella maritima*, and *Neolewinella xylanilytica* were found in saline water samples and saline-water-related environments, such as saline water sediments, saline water flora and fauna, and beach sand [3,7,8,9,10,11,43]. Concluding the analysis of known species, the number of positive samples (with species abundance of ≥0.1%) ranged from 1 to 282 samples; in addition, their maximum abundance ranged from 0.11% to 4.84% (Table 5).

Although it cannot be determined by incidence data such as in our analysis, the extensive diversity observed in aquatic and especially saline environments could be attributed to dissolved organic carbon (DOC). The amount and complexity of DOC in aquatic systems, if mirrored by diverse metabolic pathways distributed in various microorganisms, could explain some of the remarkable diversity of *Saprospirales* revealed by our analysis. Ecological reasons, like niche connectivity and increased dispersion through oceanic currents, as well as the role of oceans as the principal terminal reservoir of rainwater, may also play a role. *Saprospirales* species collected through precipitation across the land are eventually pooled in coastal seas. Currents can then contribute to the dispersal of those species across the connected oceanic bodies. Nevertheless, further investigation is needed to elucidate the observed high diversity and prevalence of *Saprospirales* in aquatic systems.

## 5. Conclusions

The introduction of next-generation sequencing allowed for high-throughput microbial profiling of environments of interest, represented, to a large extent, by the targeted sequencing of 16S rRNA gene amplicons. The relative cost efficiency and methodological simplicity led to a geometric increase in studies covering a wide range of environments around the globe. Integrating that wealth of information and reutilizing it as a unified resource for new questions is not only a very powerful approach but is also energy- and cost-effective. It is our duty as a research community to show that we can achieve the most out of the datasets we spend millions to create.

We have applied a similar integrating procedure in the past, although more demanding in execution, to the bacterial phylum *Chlamydiota* [50]. Over the years, the available datasets grow and novel databases and tools have been introduced that enable a much more streamlined querying of the global sequencing repositories. In this study, we showed that applying the tool TIC [47] over the half a million amplicon datasets in IMNGS [45] can readily give us insights into the diversity and ecological distribution of selected taxonomic groups like the order *Saprospirales*.

Future efforts should be focused on the targeted isolation of those novel members for their functional characterization and the elucidation of the ecological reasons behind that high diversity. In addition, further automatization of the pipeline used will allow further insights into other important microbial taxa that are still hindered by the limitations of isolation-based characterizations.

## Figures and Tables

**Figure 1 microorganisms-11-01767-f001:**
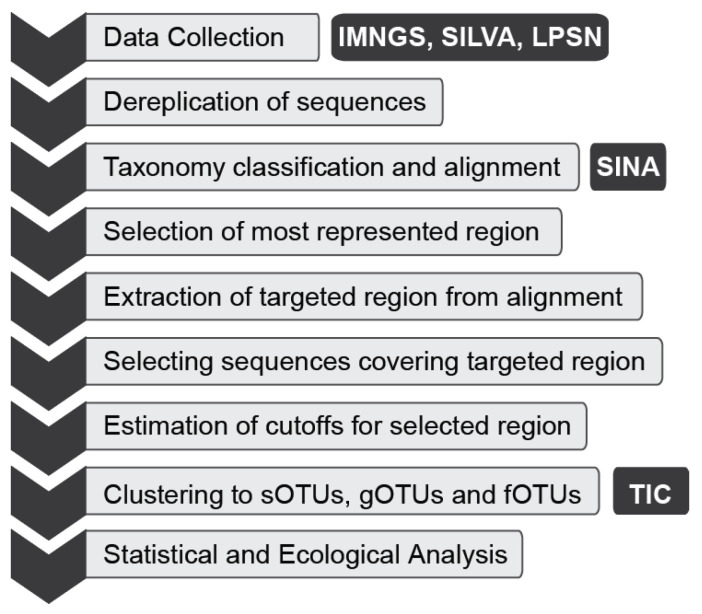
Schematic overview of the workflow followed for the diversity analysis of *Saprospirales* order.

**Figure 2 microorganisms-11-01767-f002:**
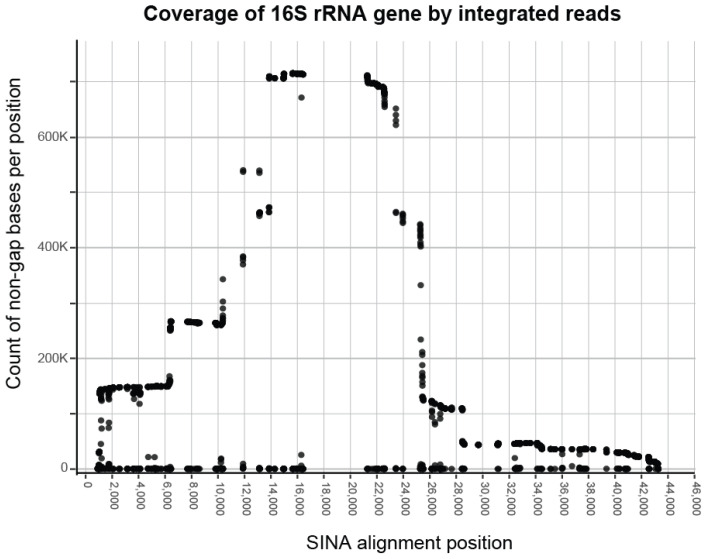
Agglomerative coverage of integrated sequences over their SINA alignment. The *y*-axis (counts) indicates the number of times a base has been found in the multiple sequence alignment for the respective position. Consecutive high counts correspond to regions overrepresented in the integrated dataset. Considering this, the most represented region spanned from the position 10 K to 25.5 K in the alignment. All aligned sequences were trimmed around these positions, and those left with a sufficient number of bases were selected for further analysis.

**Figure 3 microorganisms-11-01767-f003:**
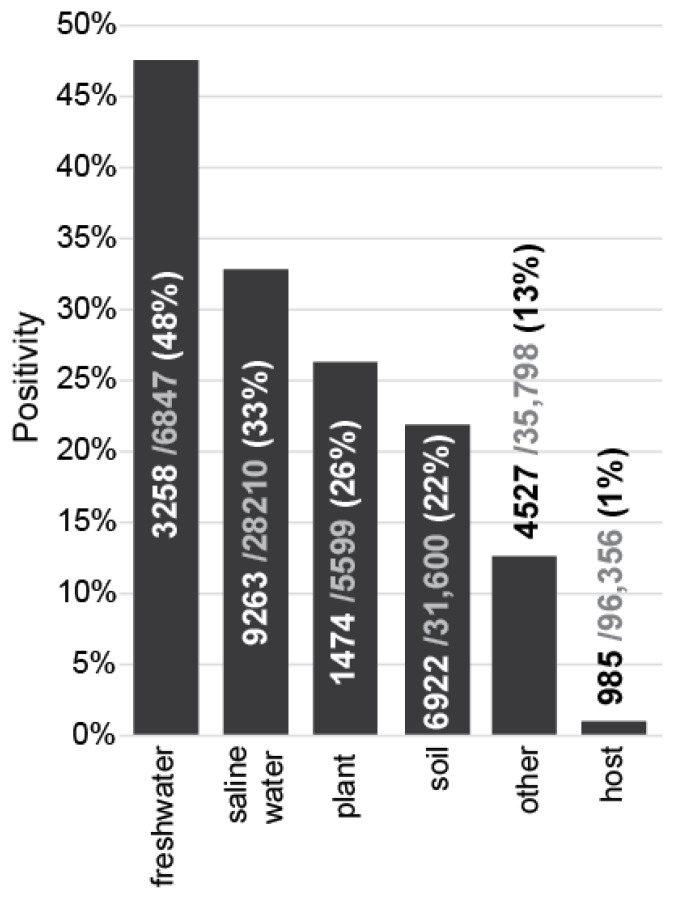
*Saprospirales* positivity (%) on IMNGS samples. The total number of samples that existed in the IMNGS database in the selected region is 204,510. These samples are categorized in the IMNGS database according to their environment of origin. In this plot, each bar represents the percentage of samples that were found to be positive for *Saprospirales* in each environmental category of IMNGS.

**Figure 4 microorganisms-11-01767-f004:**
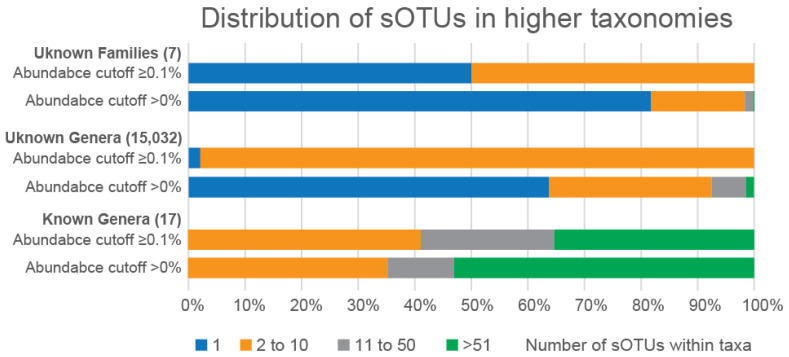
Distribution of the number of species within unknown families, unknown genera, and known genera. The three plots present the percentage of unknown families, unknown genera, and known genera that have only 1 species (blue), 2 to 10 species (orange), 11 to 50 species (grey), and more than 51 species (green).

**Figure 5 microorganisms-11-01767-f005:**
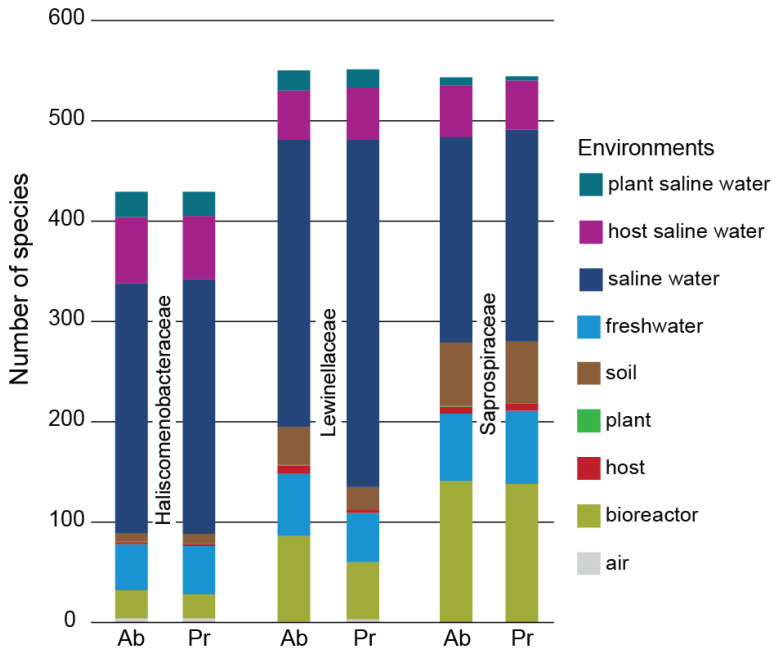
Environmental preferences by known families with species abundance ≥0.1%. Pr corresponds to prevalence (dominant environment), while Ab corresponds to maximum abundance environment. Each bar represents the number of species found in each corresponding environment.

**Figure 6 microorganisms-11-01767-f006:**
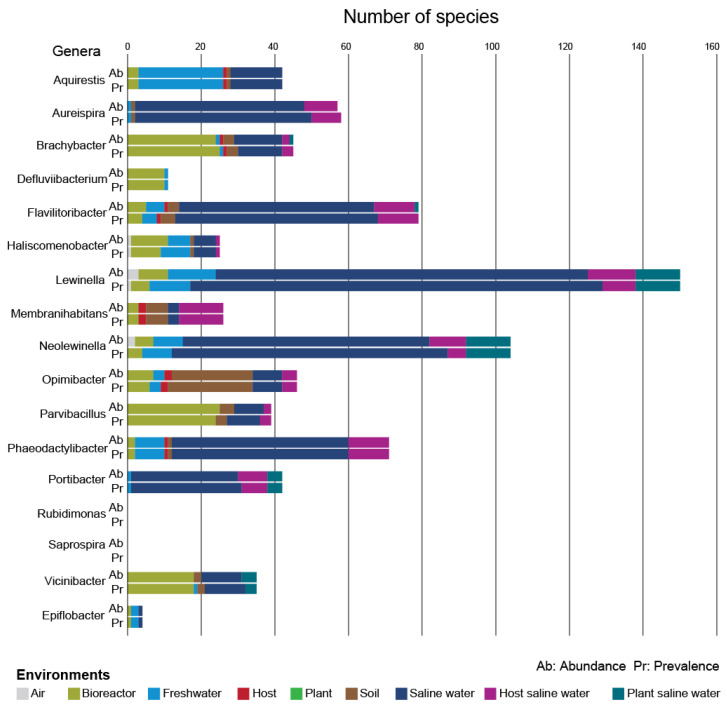
Environmental preferences by known genera with species abundance ≥0.1%. Pr corresponds to prevalence (dominant environment), while Ab corresponds to maximum abundance environments. Each bar represents the number of species found in each corresponding environment.

**Figure 7 microorganisms-11-01767-f007:**
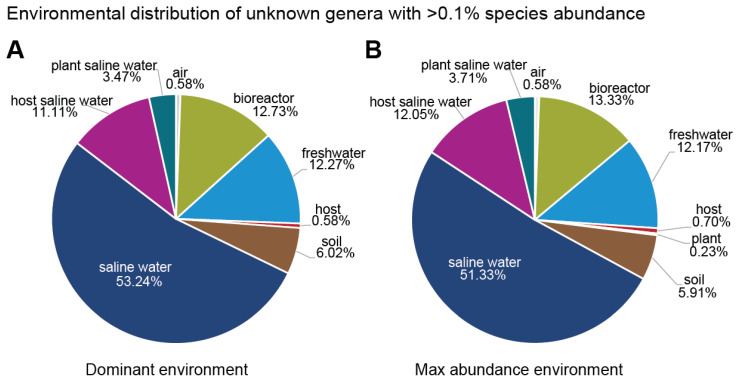
Environmental distribution of unknown genera with species abundance >0.1%. The total amount of unknown genera with species abundance >0.1% was 479 and the species belonging to them were 885. These genera belong to the three known families, as well as to the unknown families. (**A**) The pie chart presents the environmental distribution according to species prevalence. (**B**) The pie chart presents the environmental distribution according to the species’ maximum abundance environment. Each environmental percentage corresponds to the number out of the total number of species that (**A**) had their prevalence in the corresponding environment and (**B**) had their maximum abundance in the corresponding environment.

**Figure 8 microorganisms-11-01767-f008:**
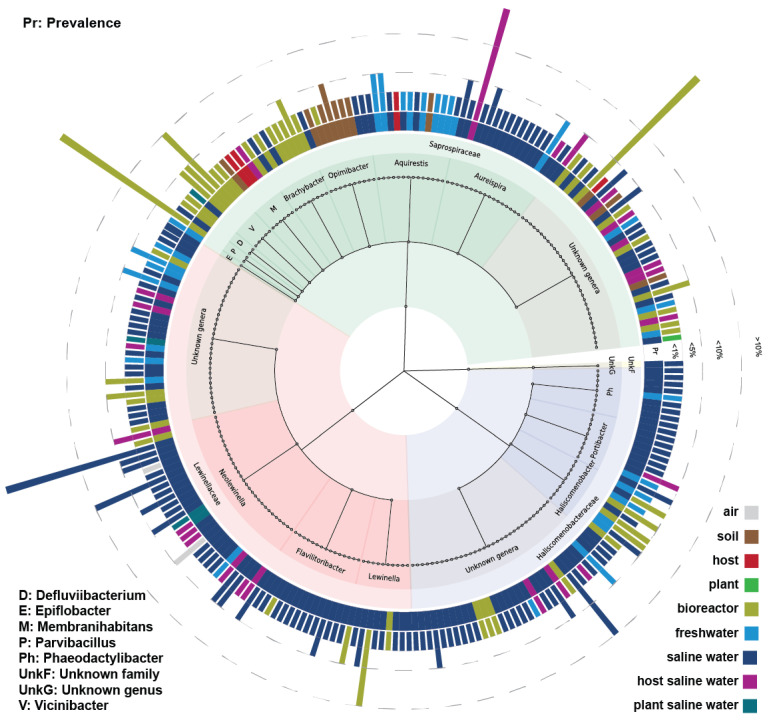
Taxonomic tree representing the environmental distribution of species appearing in at least in 50 samples each. All presented species had abundances ≥0.1%, i.e., in this diagram, 249 species out of 1565 species are presented. The diagram was created using GraPhlAn [48]. Species are presented according to the family and the genus they belong to. The inner circle of environments corresponds to species prevalence (Pr), while the rest of the circles correspond to species abundance and to the respective environment, i.e., the second circle corresponds to species abundances <1%, the third circle corresponds to abundances < 5%, and the last circle corresponds to abundances <10%.

**Figure 9 microorganisms-11-01767-f009:**
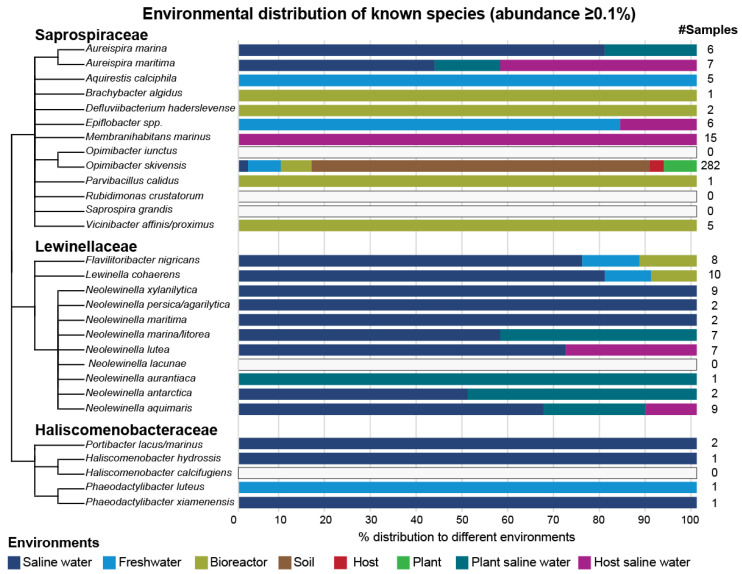
Environmental distribution (%) of known species. At the end of each bar the number of samples that these species were found in abundances of ≥0.1% are presented. *Opimibacter iunctus*, *Rubidimonas crustatorum*, *Saprospira grandis*, *Haliscomenobacter calcifugiens*, and *Neolewinella lacunae* were not found in any samples; therefore, they lack environmental distribution.

**Table 1 microorganisms-11-01767-t001:** Predicted global sequenced-based diversity of molecular families, genera, and species within the order *Saprospirales*. The last column presents the number of predicted taxa containing species with abundance ≥0.1%.

	Known	Predicted	Predicted (≥0.1%)
Families	3	1272	10
Genera	17	15,049	479
Species	32	118,062	1565

**Table 2 microorganisms-11-01767-t002:** Predicted global sequenced-based diversity of genera belonging to known and unknown families within the order *Saprospirales*.

Families	Known	Pred.	Pred. %	Pred. (≥0.1%)	Pred. % (≥0.1%)
*Saprospiraceae*	11	4483	29.79%	132	27.56%
*Lewinellaceae*	3	5430	36.08%	186	38.83%
*Haliscomenobacteraceae*	3	3675	24.42%	150	31.32%
Unknown families	NA	1461 *	9.71%	11 **	2.30%

* Distributed to 1183 unknown families, ** distributed to 7 unknown families.

**Table 3 microorganisms-11-01767-t003:** Predicted global sequenced-based diversity of species belonging to known and unknown families within the order *Saprospirales*.

Families	Known	Pred.	Pred. %	Pred. (≥0.1%)	Pred. % (≥0.1%)
*Saprospiraceae*	14	39,332	33.31%	553	35.34%
*Lewinellaceae*	13	42,921	36.36%	564	36.04%
*Haliscomenobacteraceae*	5	33,191	28.11%	435	27.76%
Unknown families	NA	2618 *	2.22%	13 **	0.83%

* Distributed to 1183 unknown families, ** distributed to 7 unknown families.

**Table 4 microorganisms-11-01767-t004:** Predicted global sequenced-based diversity of species belonging to known and unknown genera inside known families of *Saprospirales*.

Known Families	Genera	Known	Pred.	Pred. %	Pred. (≥0.1%)	Pred. % (≥0.1%)
*Saprospiraceae*	*Aquirestis*	1	3795	3.21%	42	2.68%
	*Aureispira*	2	3655	3.10%	58	3.71%
	*Brachybacter*	1	3796	3.22%	45	2.88%
	*Defluviibacterium*	1	109	0.09%	11	0.70%
	*Membranihabitans*	1	3739	3.17%	26	1.66%
	*Opimibacter*	2	1907	1.62%	47	3.00%
	*Parvibacillus*	1	2697	2.28%	43	2.75%
	*Rubidimonas*	1	4	3.4 × 10^−3^%	0	0.00%
	*Saprospira*	1	17	0.01%	0	0.00%
	*Vicinibacter*	2	1261	1.07%	35	2.24%
	*Epiflobacter*	0	34	0.03%	4	0.26%
	Unknown genera	NA	18,318 *	15.52%	242 **	15.46%
*Lewinellaceae*	*Flavilitoribacter*	1	3366	2.85%	81	5.18%
	*Lewinella*	1	3850	3.26%	46	2.94%
	*Neolewinella*	11	11,511	9.75%	105	6.71%
	Unknown genera	NA	24,194 *^3^	20.49%	332 *^4^	21.21%
*Haliscomenobacteraceae*	*Haliscomenobacter*	1	2690	2.28%	25	1.60%
	*Phaeodactylibacter*	2	2535	2.15%	71	4.54%
	*Portibacter*	2	1295	1.10%	42	2.68%
	Unknown genera	NA	26,671 *^5^	22.59%	297 *^6^	18.98%

* Distributed to 4472 unknown genera, ** distributed to 123 unknown genera, *^3^ distributed to 5427 unknown genera, *^4^ distributed to 183 unknown genera, *^5^ distributed to 3672 unknown genera, *^6^ distributed to 147 unknown genera.

**Table 5 microorganisms-11-01767-t005:** Known species assignment to sOTUs, with BLAST similarity above 98%. Each known species was assigned the ecological parameters of the respective sOTU. Symbol #, when present, stands for “The absolute number of” the corresponding entity.

Family	Genus	Species	SOTU ID	# Positive Samples (Abund. ≥ 0.1%)	Dominant Environment	Maximum Abundance Environment	Maximum Abundance
*eSaprospiraceae*	*Aquirestis*	*Aquirestis calciphila*	SOTU89	5	freshwater	freshwater	4.37%
	*Aureispira*	*Aureispira marina*	SOTU412	6	saline water	saline water	0.48%
	*Aureispira*	*Aureispira maritima*	SOTU411	7	saline water	saline water	3.41%
	*Brachybacter*	*Brachybacter algidus*	SOTU50	1	bioreactor	bioreactor	0.36%
	*Defluviibacterium*	*Defluviibacterium haderslevense*	SOTU203	2	bioreactor	bioreactor	0.56%
	*Epiflobacter*	*Epiflobacter* spp.	SOTU4123	6	freshwater	freshwater	0.56%
	*Membranihabitans*	*Membranihabitans marinus*	SOTU207	15	host saline water	host saline water	1.26%
	*Opimibacter*	*Opimibacter skivensis*	SOTU225	282	soil	soil	1.53%
	*Parvibacillus*	*Parvibacillus calidus*	SOTU68	1	bioreactor	bioreactor	0.11%
	*Vicinibacter*	*Vicinibacter affinis/proximus*	SOTU83	5	bioreactor	bioreactor	1.63%
*Lewinellaceae*	*Flavilitoribacter*	*Flavilitoribacter nigricans*	SOTU290	8	saline water	bioreactor	0.22%
	*Lewinella*	*Lewinella cohaerens*	SOTU209	10	saline water	bioreactor	1.39%
	*Neolewinella*	*Neolewinella antarctica*	SOTU384	2	saline water	plant saline water	0.88%
	*Neolewinella*	*Neolewinella aquimaris*	SOTU315	9	saline water	saline water	0.33%
	*Neolewinella*	*Neolewinella aurantiaca*	SOTU344	1	plant saline water	plant saline water	0.21%
	*Neolewinella*	*Neolewinella lutea*	SOTU309	7	saline water	saline water	4.84%
	*Neolewinella*	*Neolewinella marina/litorea*	SOTU85	7	saline water	plant saline water	0.61%
	*Neolewinella*	*Neolewinella maritima*	SOTU326	2	saline water	saline water	0.72%
	*Neolewinella*	*Neolewinella persica/agarilytica*	SOTU330	2	saline water	saline water	0.40%
	*Neolewinella*	*Neolewinella xylanilytica*	SOTU338	9	saline water	saline water	2.67%
*Haliscomenobacteraceae*	*Haliscomenobacter*	*Haliscomenobacter hydrossis*	SOTU236	1	saline water	saline water	0.91%
	*Phaeodactylibacter*	*Phaeodactylibacter luteus*	SOTU19	1	freshwater	freshwater	0.36%
	*Phaeodactylibacter*	*Phaeodactylibacter xiamenensis*	SOTU18	1	saline water	saline water	0.17%
	*Portibacter*	*Portibacter lacus/marinus*	SOTU416	2	saline water	saline water	0.17%

**Table 6 microorganisms-11-01767-t006:** The 20 most prevalent species accompanied by their respective information regarding the environment of prevalence, the maximum abundance environment and the corresponding maximum abundance, and the number of samples identified. The sOTUs with BLAST similarity above 98% were assigned to known species. Symbol #, when present, stands for “The absolute number of” the corresponding entity.

Family	Genus	Species	SOTU ID	# Positive Samples	# Positive Samples (Abundance ≥ 0.1%)	Environment of Prevalence	Max Abundance Environment	Max Abundance
*Saprospiraceae*	*Opimibacter*	*Opimibacter skivensis*	SOTU225	3066	282	soil	soil	1.53%
*Lewinellaceae*	*Neolewinella*	Unknown species	SOTU306	516	25	saline water	saline water	12.20%
*Lewinellaceae*	*Neolewinella*	Unknown species	SOTU307	420	20	saline water	saline water	1.82%
*Lewinellaceae*	*Flavilitoribacter*	Unknown species	SOTU4353	415	27	saline water	saline water	2.52%
*Lewinellaceae*	*Neolewinella*	Unknown species	SOTU321	408	9	saline water	saline water	1.43%
*Saprospiraceae*	*Aureispira*	Unknown species	SOTU414	333	20	saline water	saline water	0.90%
*Lewinellaceae*	*Neolewinella*	Unknown species	SOTU3384	332	31	saline water	saline water	1.03%
*Saprospiraceae*	GOTU74	Unknown species	SOTU5520	327	14	saline water	saline water	2.43%
*Lewinellaceae*	*Flavilitoribacter*	*Flavilitoribacter nigricans*	SOTU290	315	8	saline water	bioreactor	0.22%
*Saprospiraceae*	*Aquirestis*	Unknown species	SOTU112	287	11	freshwater	saline water	0.31%
*Haliscomenobacteraceae*	*Haliscomenobacter*	Unknown species	SOTU277	285	21	freshwater	freshwater	0.45%
*Saprospiraceae*	GOTU946	Unknown species	SOTU527	272	12	soil	soil	0.42%
*Haliscomenobacteraceae*	*Phaeodactylibacter*	Unknown species	SOTU11811	265	12	saline water	saline water	0.71%
*Lewinellaceae*	*Neolewinella*	Unknown species	SOTU310	263	13	saline water	air	0.39%
*Lewinellaceae*	*Neolewinella*	*Neolewinella aquimaris*	SOTU315	256	9	saline water	saline water	0.33%
*Haliscomenobacteraceae*	GOTU588	Unknown species	SOTU4189	253	3	saline water	saline water	0.41%
*Saprospiraceae*	*Aquirestis*	Unknown species	SOTU187	246	2	saline water	freshwater	0.36%
*Lewinellaceae*	*Flavilitoribacter*	Unknown species	SOTU1854	243	10	saline water	saline water	0.35%
*Lewinellaceae*	*Lewinella*	Unknown species	SOTU220	234	4	saline water	bioreactor	7.92%
*Haliscomenobacteraceae*	*Portibacter*	Unknown species	SOTU10607	233	2	saline water	saline water	1.62%

**Table 7 microorganisms-11-01767-t007:** The 20 most abundant species accompanied by their respective information regarding the environment of prevalence, the maximum abundance environment and the corresponding maximum abundance, and the number of samples identified. The sOTUs with BLAST similarity above 98% were assigned to known species. Symbol #, when present, stands for “The absolute number of” the corresponding entity.

Family	Genus	Species	SOTU ID	# Positive Samples	# Positive Samples (Abundance ≥ 0.1%)	Environment of Prevalence	Max. Abundance Environment	Max. Abundance
*Saprospiraceae*	*Parvibacillus*	Unknown species	SOTU51	230	25	bioreactor	bioreactor	28.02%
*Saprospiraceae*	*Aureispira*	Unknown species	SOTU413	71	1	host saline water	host saline water	22.60%
*Saprospiraceae*	GOTU1969	Unknown species	SOTU83565	1	1	bioreactor	bioreactor	21.73%
*Saprospiraceae*	*Parvibacillus*	Unknown species	SOTU80	2	1	bioreactor	bioreactor	14.97%
*Lewinellaceae*	GOTU1025	Unknown species	SOTU13902	27	1	saline water	saline water	12.90%
*Saprospiraceae*	GOTU42	Unknown species	SOTU565	205	14	bioreactor	bioreactor	12.72%
*Lewinellaceae*	*Neolewinella*	Unknown species	SOTU306	516	25	saline water	saline water	12.20%
*Haliscomenobacteraceae*	*Phaeodactylibacter*	Unknown species	SOTU6	5	3	saline water	saline water	11.92%
*Saprospiraceae*	GOTU741	Unknown species	SOTU7213	49	8	saline water	saline water	11.79%
*Saprospiraceae*	GOTU4577	Unknown species	SOTU50336	6	1	bioreactor	bioreactor	10.01%
*Saprospiraceae*	*Parvibacillus*	Unknown species	SOTU57	11	2	bioreactor	bioreactor	9.33%
*Lewinellaceae*	*Neolewinella*	Unknown species	SOTU312	105	3	saline water	saline water	9.23%
*Lewinellaceae*	GOTU755	Unknown species	SOTU14539	7	3	bioreactor	bioreactor	8.93%
*Lewinellaceae*	*Lewinella*	Unknown species	SOTU220	234	4	saline water	bioreactor	7.92%
*Haliscomenobacteraceae*	GOTU423	Unknown species	SOTU42257	12	3	saline water	saline water	7.45%
*Lewinellaceae*	*Neolewinella*	Unknown species	SOTU14993	21	1	freshwater	freshwater	7.37%
*Saprospiraceae*	*Brachybacter*	Unknown species	SOTU31	32	7	bioreactor	bioreactor	7.31%
*Haliscomenobacteraceae*	GOTU2075	Unknown species	SOTU32764	12	2	freshwater	bioreactor	7.12%
*Lewinellaceae*	GOTU6152	Unknown species	SOTU72597	4	1	bioreactor	bioreactor	6.24%
*Haliscomenobacteraceae*	GOTU62	Unknown species	SOTU4917	124	3	saline water	saline water	5.89%

## Data Availability

The data presented in this study are openly available at https://www.imngs.org/ and https://www.arb-silva.de/ (accessed on October 2022).

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
