# Peer review of "Large-Scale Integration of Amplicon Data Reveals Massive Diversity within Saprospirales, Mostly Originating from Saline Environments"

_microorganisms, 2023, doi:10.3390/microorganisms11071767_

Round 1

Reviewer 1 Report

The authors evaluated the taxa diversity of order Saprospirales in public NGS database based on 16S rRNA similarities around the selected SINA positions. And 1183 novel families, 15033 novel genera, and 188K novel species were detected. Lots of work need to be done to see if those results hold true. Below are some detail questions about this paper:

1.      Almost all the novel taxa in this study were detected by environmental dataset (eDNA sequences), it’s hard to identify new species without information of isolated strains, such as morphological features, full-length 16S rRNA gene sequences, phylogenetic relationship with relatives, etc.

2.      All newly discovered species mentioned in this study are only based on a possibility. Therefore, when discussing them, it is important to be cautious with the language used and to apply strict detection thresholds.

3.      In line 112-119, only 31 reference sequences were used to draw the conclusion that the appropriate thresholds of 16S rRNA gene similarity at species, genera and family levels were 96%, 92% and 90% respectively. The results are a bit too hasty. First, 31 sequences were too few. Second, how about non-Saprospirales taxa? Are there any non-Saprospirales taxa shared >96%, >92% or 90% similarity to these 31 reference sequences? 90% and 92% similarity thresholds are very loose thresholds. All environmental sequences should be matched with public database first to confirm that they belong to the order Saprospirales, or at least demonstrate that they are not sequences from other known orders, before proceeding with further analysis.

4.      The authors need to first demonstrate the effectiveness of their method for detecting new taxa before evaluating the abundance and proportion of corresponding species in the environment.

Author Response

We thank reviewer 1 for his thorough review of our work and his insightful comments. Please read our detailed responses on each comment below.

  1. Almost all the novel taxa in this study were detected by environmental dataset (eDNA sequences), it’s hard to identify new species without information of isolated strains, such as morphological features, full-length 16S rRNA gene sequences, phylogenetic relationship with relatives, etc.

We completely agree that the identification of new species is a laborious process. The requirement for the official description of species includes a pure culture and it is one of the biggest limiting factors in expanding the base of named species. Nevertheless, we can now utilize metagenomic techniques to explore the existence of microorganisms (as we do) or obtain functional insights (Genomes from metagenomic sequencing) on so far uncultured taxa. Our work highlights the extent of further work and attention needed in the actual isolation and characterization of the massive still unknown members of Saprospirales while pointing out specific environments where researchers can focus their efforts on isolating the biological species carrying the 16S rRNA gene sequences determined by our analysis

  1. All newly discovered species mentioned in this study are only based on a possibility. Therefore, when discussing them, it is important to be cautious with the language used and to apply strict detection thresholds.

We agree that biological species are different from molecular species. Given that the definition of biological microbial species is at best problematic and rather a human construct for communication and knowledge organization we agree with the reviewer’s comment that clusters of sequences sharing a certain similarity cannot be called species in the classical sense. We modify the text, to the extent of not hindering readability, to add the distinction of molecular species, genera and families in front of clusters of the corresponding similarity. Nevertheless, we feel that we have to add that the presence of the phylotypes described in our work is more than a possibility. Those sequences went through rigorous quality and chimera filtering. They represent pieces of evidence for the existence of real organisms. 

  1. In line 112-119, only 31 reference sequences were used to draw the conclusion that the appropriate thresholds of 16S rRNA gene similarity at species, genera and family levels were 96%, 92% and 90% respectively. The results are a bit too hasty. First, 31 sequences were too few. Second, how about non-Saprospirales taxa? Are there any non-Saprospirales taxa shared >96%, >92% or 90% similarity to these 31 reference sequences? 90% and 92% similarity thresholds are very loose thresholds. All environmental sequences should be matched with public database first to confirm that they belong to the order Saprospirales, or at least demonstrate that they are not sequences from other known orders, before proceeding with further analysis.

In line 112-119, only 31 reference sequences were used to draw the conclusion that the appropriate thresholds of 16S rRNA gene similarity at species, genera and family levels were 96%, 92% and 90% respectively. The results are a bit too hasty. First, 31 sequences were too few. Second, how about non-Saprospirales taxa

It is important to adjust our similarity cutoffs to the selected amplicon region, as it is clear that fragments of the 16S rRNA gene do not follow the same similarity as the whole gene.  We went to great lengths to ensure we collect all known Saprospirales. There are no other species we are aware of besides those 31 we included. We agree that for a whole order to only have 31 species is a very low number and it justifies our effort to explore further the diversity of this order. Nevertheless, we do not agree that adding non-Saprospirales sequences to our dataset used to calibrate the taxonomic borders within Saprospirales is justified. We understand the concern but if we have chosen your recommended approach, we again should justify why we believe that the resulted ranges coming for a wide selection of bacteria are still relevant for Saprospirales. It is not uncommon to see taxa to be broken into multiple subdivisions as they do not obey the similarity limits regularly used for those taxonomic levels. Nonetheless, we admit that this idea needs to be tested further. It would simplify future projects if a universal dataset can be used to adjust similarity cutoffs for different regions of the 16S rRNA gene.

Are there any non-Saprospirales taxa shared >96%, >92% or 90% similarity to these 31 reference sequences? 90% and 92% similarity thresholds are very loose thresholds.

All of the 691K sequences within our dataset have been classified first into Saprospirales by SILVA. It means that ALL of them fall within a pure Saprospirales clade that allows the SINA classifier to assign them confidently to the order level.  They might be non-Saprospirales sequences outside of our dataset that have similarities over 90% to our sequences but they classify as something else. The misleading signal of naïve similarity is the leading reason for the usage of TIC tool in our analysis as it applies all taxonomic/phylogenetic information first and then performs bounded similarity-based incremental clustering.  We do not understand in what sense 90% is a loose threshold for dividing novel 16S rRNA gene sequences, known to belong in the same order, to different clusters equivalent to the family taxonomic level within that order. The standard similarity cutoff for families is 90% similarity, meaning that a new isolate with less than 90% similarity to the sequences of known species probably belongs to a different family. We are aware that phylogeny and other parameters are needed for the official declaration of a new family but the similarity cutoff of 90% is a good first step. For genera, our 92% regional similarity cutoff is actually stricter than the default 95% as it reduces the number of novel divisions. Sequences that only show similarity of 93% to known sequences for example will still be included in the same “genus” cluster, while if the 95% cutoff was used would have formed a new genus-level cluster. In fact, all our thresholds are stricter than the defaults at the corresponding level in order to avoid inflation of novel taxa. 

All environmental sequences should be matched with public database first to confirm that they belong to the order Saprospirales, or at least demonstrate that they are not sequences from other known orders, before proceeding with further analysis.

As we write in our answer above, all 691K sequences in the analysis were classified confidently to Saprospirales with SINA. Taxonomic classification based on phylogenetic informative positions on a sequence (like SINA or RDP classifier) has been shown to outperform classifications relying on naïve similarity matching like BLAST. We extensively modify the methods’ text to clarify all filtering steps performed in our analysis. We agree that it is important that the fact that our sequences are pure Saprospirales beyond reasonable questioning should be clear and transparent. We are open to suggestions on points where additional clarity is needed.

  1. The authors need to first demonstrate the effectiveness of their method for detecting new taxa before evaluating the abundance and proportion of corresponding species in the environment.

We are not sure which part of the method the reviewer questions referring to with “effectiveness for detecting new taxa”. The use of sequences obtained from metagenomic experiments to detect unknown organisms is a common practice. We assume that sequencing plasmid libraries with 16S rRNA targeting PCR products in the past with Sanger sequencing and new NGS community profiling nowadays are not in question of molecular methods for detecting new taxa. Since detection is different from proving usually additional steps were followed to prove the new taxons existence. Usually, the first step is to design specific probes or primers based on those sequences and look at the environment of origin with methods like FISH or specific PCR and re-sequencing. We could not do that because we do not have access to the original samples but we also do not feel we need to do that as it has been done extensively in other studies. Therefore, we assume that the question refers to how our TIC-based clustering compares to naïve similarity-based clustering.  We have shown that on the original publication of TIC. In the original publication, we show that TIC is superior to naïve similarity clustering and should be the method of choice in microbial community or diversity studies. 

Author Response

We are grateful to reviewer 2 for the insightful comments showing the understanding and appreciation of the methodological advancements our work represents.  

Comment 1: The only suggestion would be to add more information about the application of this tool (like TIC) in other organisms for diversity studies

We are glad for this comment highlighting the community’s interest in exploring the sequence-based diversity of other taxonomic groups. Although any group that includes a bioinformatician can easily follow our methodology on other taxa, we admit that biologists will find this process demanding without any programming knowledge. We are working on a fully automated online service, based on the methodology of this manuscript, that would allow anyone to easily explore the diversity of any taxon through an intuitive user interface. We revise and extend the method section to include additional information when needed, although custom scripts are still required to perform all steps.  Groups interested in similar studies can always contact us for support until we release the automated pipeline.

Comment 2: The author perspectives on the reasons behind why Sapropriages were most prevalent in saline water would make a more perfect discussion.

We agree with the reviewer’s comment about the need for some perspective on Saprospirales observed diversity and prevalence patterns.  Since this is a descriptive study, we can only offer some hypotheses that need to be evaluated by actual experimental work. We add one whole paragraph in the discussion with our explanation of why oceans act as a reservoir for Saprospirales.

We add the following “Although it cannot be determined by incidence data like our analysis, the extensive diversity observed in aquatic and especially saline environments could be attributed to Dissolved Organic Carbon (DOC). The amount and complexity of DOC in aquatic systems, if mirrored by diverse metabolic pathways distributed in various microorganisms could explain some of the remarkable diversity of Saprospirales revealed by our analysis. Ecological reasons, like niche connectivity and increased dispersion through oceanic currents as well as the role of oceans as the principal terminal reservoir of rainwater, may also play a role. Saprospirales species collected through precipitation across the land are eventually pooled in coastal seas. Currents can then contribute to the dispersal of those species across the connected oceanic bodies. Nevertheless. further investigation is needed to elucidate the observed high diversity and prevalence of Saprospirales in aquatic systems.”

Reviewer 3 Report

This is a very good paper.I only have two minor requirements:

With line 72 and “Databases, such as IMNGS, contain 16S rRNA microbial profiles from more than 500,000 preprocessed samples across the globe.” I suggest you add a reference: Sandle, T. (2019) Application of a Riboprinter® for Microbiological Control in Pharmaceuticals, Journal of GxP Compliance, 23 (6): http://www.ivtnetwork.com/article/application-riboprinter%C2%AE-microbiological-control-pharmaceuticals

With Line 89,it is important to emphasize that ‘Taxonomy Informed Clustering’ is novel tool.

Author Response

We like to thank reviewer 3 for his time to review our manuscript and his suggestions.

Comment 1: With line 72 and “Databases, such as IMNGS, contain 16S rRNA microbial profiles from more than 500,000 preprocessed samples across the globe.” I suggest you add a reference: Sandle, T. (2019) Application of a Riboprinter® for Microbiological Control in Pharmaceuticals, Journal of GxP Compliance, 23 (6): http://www.ivtnetwork.com/article/application-riboprinter%C2%AE-microbiological-control-pharmaceuticals

We are open to suggestions of additional literature references that would enhance readers’ understanding of our methodology or the appropriate contextualization of our results. Nevertheless, in our evaluation, the suggested reference is not relevant to the context of the sentence in question.  We welcome alternative literature suggestions or the context in which the particular citation is relevant.

Comment 2: With Line 89, it is important to emphasize that ‘Taxonomy Informed Clustering’ is a novel tool.

We agree with the comment of reviewer 3 about the importance of additional emphasis on the novelty of the tool “Taxonomy Informed Clustering” we used for our analysis. We extensively modify, through rephrasing, our method section and often added the word “novel” when referring to TIC methodology plus a new sentence briefly explaining the operation of the TIC algorithm as: “TIC is a new clustering algorithm that first procedurally divides taxonomically annotated sequences into bins of the same taxonomy down to genus level. Then performs incremental clustering using the sequences confined within the same taxonomy level to avoid contamination of the clusters with sequences with clearly different phylogenetic origin but otherwise overall sequence similarity above the set cutoff levels.”.

Round 2

Reviewer 1 Report

I have no question now.